

# Random forest meteorological normalisation models for Swiss $PM_{10}$ trend analysis

Stuart K. Grange[1,2], David C. Carslaw[1,4], Alastair C. Lewis[1,3], Eirini Boleti[2,5], and Christoph Hueglin[2]

[1]Wolfson Atmospheric Chemistry Laboratories, University of York, York, YO10 5DD, United Kingdom
[2]Empa, Swiss Federal Laboratories for Materials Science and Technology, 8600 Dübendorf, Switzerland
[3]National Centre for Atmospheric Science, University of York, Heslington, York, YO10 5DD, United Kingdom
[4]Ricardo Energy & Environment, Harwell, Oxfordshire, OX11 0QR, United Kingdom
[5]EPFL, École Polytechnique Fédérale de Lausanne, Route Cantonale, 1015 Lausanne, Switzerland

*Correspondence to:* Stuart K. Grange (stuart.grange@york.ac.uk)

**Abstract.**

Meteorological normalisation is a technique which accounts for changes in meteorology over time in an air quality time series. Controlling for such changes helps support robust trend analysis because there is more certainty that the observed trends are due to changes in emissions or chemistry, not changes in meteorology. Predictive random forest models (RF; a decision

tree machine learning technique) were grown for 31 air quality monitoring sites in Switzerland using surface meteorological, synoptic scale, boundary layer height, and time variables to explain daily $PM_{10}$ concentrations. The RF models were used to calculate meteorologically normalised trends which were formally tested and evaluated using the Theil-Sen estimator. Between 1997 and 2016, significantly decreasing normalised $PM_{10}$ trends ranged between -0.09 and -1.16 $\mu g \, m^{-3} \, year^{-1}$ with urban traffic sites experiencing the greatest mean decrease in $PM_{10}$ concentrations at -0.77 $\mu g \, m^{-3} \, year^{-1}$. Similar magnitudes have

been reported for normalised $PM_{10}$ trends for earlier time periods in Switzerland which indicates $PM_{10}$ concentrations are continuing to decrease at similar rates as in the past. The ability for RF models to be interpreted was leveraged using partial dependence plots to explain the observed trends and relevant physical and chemical processes influencing $PM_{10}$ concentrations. Notably, two regimes were suggested by the models which cause elevated $PM_{10}$ concentrations in Switzerland: one related to poor dispersion conditions and a second resulting from high rates of secondary PM generation in deep, photochemically active

boundary layers. The RF meteorological normalisation process was found to be robust, user friendly and simple to implement, and readily interpretable which suggests the technique could be useful in many air quality exploratory data analysis situations.

## 1 Introduction

### 1.1 Air quality trend analysis

Trend analysis of ambient air quality data is a common and important procedure. The goal of such trend analysis usually

involves the confirmation, or lack of confirmation of a statistically significant change in pollutant concentrations over time. If pollutant concentrations are significantly increasing or decreasing, there is evidence that air quality is better or worse than in the past and conclusions such as these are useful for scientists, policy makers, and the public (Porter et al., 2001). However,



air quality trend analysis is complicated because it is usually unknown if the behaviour of the trend is driven by changes in meteorology or changes in emissions or atmospheric chemistry (Rao and Zurbenko, 1994; Pryor et al., 1995; Libiseller and Grimvall, 2003; Libiseller et al., 2005; Wise and Comrie, 2005). The former is usually is of greatest importance for policy makers because investigation in changes in emissions, and in turn, the perturbations on ambient pollutant concentrations is
how efficacy of intervention activities are judged (Zeldin and Meisel, 1978; Carslaw et al., 2006). Despite the uncertainty surrounding the drivers of air pollutant trends, this issue is often acknowledged but rarely robustly compensated for.

    The issue surrounding meteorology and air quality trend analysis arises because air quality and pollutant concentrations are highly dependent on meteorological conditions across all spatial scales (Stull, 1988). Wind speed, wind direction, atmospheric temperature and stability can be expected to have large influences on pollutant concentrations at most locations. The influence
of such meteorological variables can be much greater than an intervention activity which results in meteorological conditions often obscuring or exacerbating trends (Anh et al., 1997). In situations where these processes are not accounted for, a calculated trend is less likely to represent changes in pollutant emissions due to air quality management efforts and therefore erroneous conclusions can be made on what is causing the observed trend.

    The methods used for trend analysis are diverse and range from simple least squares linear regression analysis to numeri-
cally complex methods often requiring multiple pre-processing or work-up steps before the final trend test is conducted (Lou Thompson et al., 2001; Porter et al., 2001; Marchetto et al., 2013). When trends are found to be monotonic, *i.e.* constantly changing with time, the robust non-parametric linear regression Mann-Kendal test is often used (Guerreiro et al., 2014). The Mann-Kendal test can be supplemented by using the Theil-Sen estimator and bootstrapping techniques which increase the test's robustness and can account for autocorrelation in the time series (Siegel, 1982; Hamed and Ramachandra Rao, 1998; Salmi
et al., 2002). Although methods for the testing of monotonic trends are mature and are in common usage in air quality and other environmental applications (Meals et al., 2011), much of the effort of trend analysis is put into the pre-processing steps which generally involves deciding what aggregation period and function to use as well as handling the removal of the seasonal component if necessary (an annual cyclical pattern). Common techniques to allow for removal of the seasonal component of a time series is classical decomposition using loess (often called seasonal and trend decomposition using loess; STL) (Cleveland
et al., 1990) and Kolmogorov-Zurbenko filters (Wise and Comrie, 2005; Yang and Zurbenko, 2010). Although these decomposition methods help treat the time series for further trend analysis, they alone do not address changes of meteorology over time.

### 1.2   Meteorological normalisation

A method to control or take into account meteorology effects on pollutant concentrations involves the development and use of
predictive statistical models (Lou Thompson et al., 2001; Carslaw et al., 2006; Beevers et al., 2009; Carslaw and Priestman, 2015; Fuller and Carslaw, 2017). Such models attempt to use a number of explanatory variables such as surface measurements of wind behaviour, atmospheric temperature, and pressure to explain the variability of pollutant concentrations. Time variables such as Julian day (day of the year), weekday, and hour of the day can also be used as predictors. These time variables act as proxies for emission strength because pollutant emissions or generation processes vary by the time of day, day of the week, and





season (Derwent et al., 1995). If the predictive models are found to explain an adequate amount of the variation in pollutant concentration, the model can be used to account for the influence of meteorological variables on the pollutant concentration. The explanation of some of the variation in a time series also has the side effect of allowing significant trends to be detected earlier because of the reduction of estimate uncertainty. This technique is known by a few different names but here, we refer to

the technique as *meteorological normalisation*.

The application of meteorological normalisation approaches are however complicated due to how pollutant concentrations vary based on meteorological variables. For example, for a traffic sourced pollutant such as nitrogen dioxide ($NO_2$), it would be expected that concentrations would decrease with increasing wind speed due to atmospheric dilution and dispersion processes (Hitchins et al., 2000). However, this process is highly unlikely to be linear and when a monitoring site is located adjacent

to a kerb, the effect of dilution based on the wind speed would also be highly dependent on wind direction. There would be further complication if the monitoring site was located within a street canyon. When variables depend on one another (or among more than two variables) in such a way, this is termed interaction (Cox, 1984). Interaction effects generally require special treatment in most statistical models. Additionally normality, homoscedasticity, multicollinearity, and independence should also be addressed before and during statistical modelling. All of these features are commonly encountered in air quality

time series which can make the use statistical techniques highly burdensome in this domain.

## 1.3   Machine learning

In the past three decades, there has been large development in the field what is now known as machine learning (ML). ML is a fusion of statistics, data science, and computing which experiences use across a very wide range of applications (Smola and Vishwanathan, 2008; Kuhn, 2013). ML is a diverse topic but it has seen the development of many predictive models

which offer alternatives to "classical" statistical models for exploratory data analysis. Some of the more popular ML predictive models include decision tree methods such as boosted regression trees and random forest, the kernel methods which include support vector machines, and finally artificial neural networks (Friedman, 2006). These ML methods, when used in regression mode, can be used in similar applications as multiple regression models such as general additive models (GAMs). These ML techniques are non-parametric and have the critical advantage of not needing to address many of the assumptions needed for

statistical models such as sample normality, homoscedasticity, independence, adherence to other strict parametric assumptions, and the careful handling of interaction effects (Immitzer et al., 2012). ML predictive models have the potential to supplement more classical statistical techniques which may result in improved air quality trend analysis.

### 1.3.1   Decision trees and random forest

Random forest (RF) (also known as decision forests) which is utilised in this study is an ensemble decision tree ML method

(Breiman, 2001; Tong et al., 2003). Decision trees use a binary recursive classifying algorithm which creates "pure" nodes by splitting observations into two homologous groups. The recursive nature of the algorithm means splitting is repeated until node purity is achieved. Together the entire series of splits, individually called nodes or branches, is referred to as a tree. The recursive algorithm will always correctly classify input data if the trees are allowed to grow to their maximum depth.





Algorithms of this sort are called greedy (Biau et al., 2008). This greedy behaviour can result in very deep trees (especially with continuous numeric variables) where the final split is only evaluating two observations *i.e.*, a singleton node. Models such these will very rarely generalise to new data which was not used to train the model. Therefore, decision trees are prone to overfitting (Kotsiantis, 2013). RF controls for this disadvantage by growing many individual decision trees from a training

set using a process called bagging (bootstrap aggregation). RF is an ensemble method because the model consists of many individual trees/models/learners grown from bagged data but when used for prediction, all the trees' outputs are used together (Figure 1).

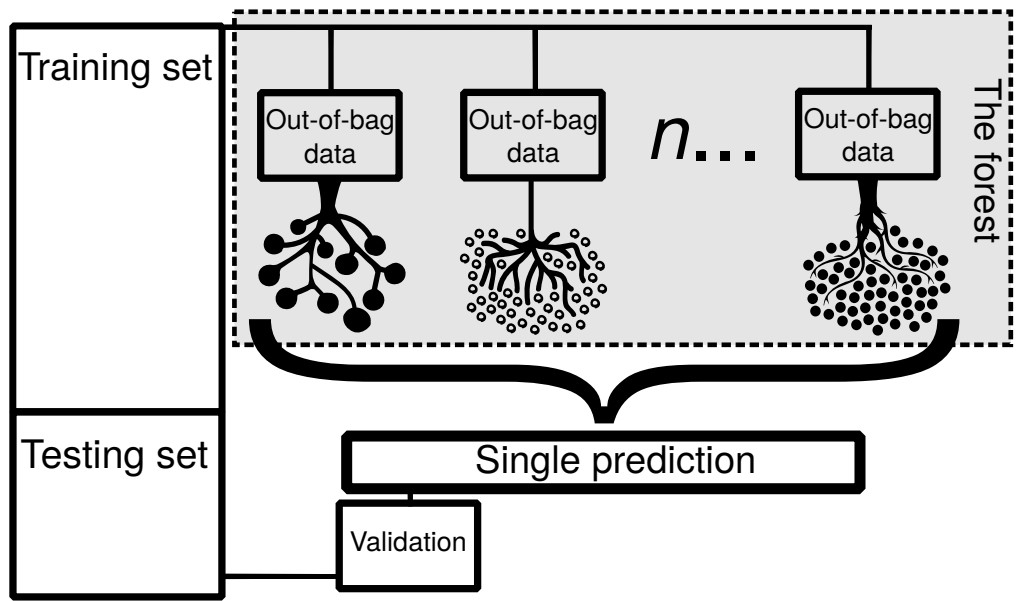

**Figure 1.** Conceptual diagram of a random forest model. Many out-of-bag samples are taken from the training set and different decision trees are grown. After many decision trees are grown, termed the forest, all trees are used to form a single prediction. The predictions can then be validated using the test set which is withheld from the training process. Tree icons are from freepik.com (2017).

Bagging refers to randomly sampling observations with replacement from the training set along with sampling of explanatory variables (Breiman, 1996). A set which results from bagging is called out-of-bag data (OOB) and OOB data will always be

lacking some of the input data. When a single tree is grown from OOB data, it is unlikely to contain the same observations and variables used by other trees if the process is repeated. RF models usually contain a few hundred trees using OOB data and this creates a forest which consists of many decorrelated trees which have been trained on different subsets of the training set (Figure 1). Every tree can then be used to predict and the predictions are aggregated to form a single prediction. In regression applications, the mean of predictions is used. Somewhat counter intuitively, the bagging process and ensemble predictions

addresses decision trees' tendency to overfit training sets (Friedman et al., 2001). This allows RF to produce predictive models which generalise well and predictive performance is generally considered among the best of any ML technique (Caruana and Niculescu-Mizil, 2006).





RF also has the advantage of not being a "black-box" method (Jones and Linder, 2015). Decision trees are one of the few ML techniques where the learning process can be explained, investigated, and interpreted. In the case of artificial neural networks or kernel based learning methods, this is much more difficult to do (Kotsiantis, 2013; Tong et al., 2003). RF models can be investigated with partial dependence plots which demonstrate the relationships among variables and a variable's importance

as a predictor can be determined. RF can be used in unsupervised, regression, or classification modes, accepts numeric and categorical variables, and is known to be simpler to tune when compared to other decision tree methods which usually require pruning; a process which removes some of the grown branches from the forest. The combination of these attributes has made RF a popular ML technique (Friedman et al., 2001; Immitzer et al., 2012).

### 1.4 Objectives

Improvements in the pre-processing steps for air quality trend analysis need to be made which control, or account for meteorology and allow for more robust trend and intervention exploration. This paper has the overall objective to present a meteorological normalisation technique which uses RF predictive models to prepare ambient atmospheric pollutant concentration data for trend analysis. Specifically, this paper will (*i*) present a meteorological normalisation technique using RF predictive models using routine data which will be accessible to most data users, (*ii*) present a trend analysis of the meteorologically normalised

time series, and (*iii*) use RF's advantage of being able to interpret the learning processes to explain the trends which are observed. Daily $PM_{10}$ observations from across Switzerland will be used for the analysis. The use of daily Swiss $PM_{10}$ data was chosen because the data record and capture rates are excellent, and a previous study (Barmpadimos et al., 2011) conducted a $PM_{10}$ trend analysis using a different method for observations between 1991 and 2008. Therefore, this work also updates and extends previous work.

## 2 Methods

### 2.1 Data

Routine air quality observations from Switzerland were used in this study and these data were accessed from the European Environment Agency (EEA) AirBase and Air Quality e-Reporting (AQER) data repositories (European Environment Agency, 2014, 2017). The AirBase repository includes data between 1969 and 2012 (inclusive) while the AQER repository contains

25 data from 2013 onwards. These two repositories contain monitoring sites which are within Switzerland's National Air Pollution Monitoring Network (NABEL) and sites which are managed by the Swiss Cantons (states) (Federal Office for the Environment, 2014, 2017). Data from the two repositories have different data models and file formats which required transformation and processing into a standardised relational data model called **smonitor** (Grange, 2016, 2017). The Härkingen-A1 and Sion-Aéroport sites' data are not submitted to the EEA, therefore these data were requested and delivered directly from the Swiss

Federal Office for the Environment (FOEN).



Daily $PM_{10}$ observations were used as the pollutant of interest and in the models as the dependent variable. Observations between 1997 and 2016 were used and the observations were collected with the use of commercially available gravimetric instrumentation and are subjected to quality assurance and control procedures (Federal Office for the Environment, 2017). A total of $186\,400$ $PM_{10}$ observations from 31 sites were used. The sites were classified into six site types: rural, rural mountain, urban background, suburban, rural motorway, or urban traffic based on classifications in the AQER reporting system. For site locations and details see Table 1 and Figure 2.

The 31 $PM_{10}$ monitoring sites where chosen for their suitably for use in trend analysis. The main condition was that $PM_{10}$ observations needed to be unbroken for at least five years. One exception was made for Zürich-Schimmelstrasse. Zürich-Schimmelstrasse has a broken $PM_{10}$ time series due to $PM_{10}$ monitoring occurring every second year between 2002 and 2010, however, these data were still considered valuable to include in the analysis. All other sites had very high data capture rates (median of 99 %) for the duration they were operational. Five monitoring sites were closed before, or did not have $PM_{10}$ data to the end of the analysed time period (the end of 2016) but until their date of closure, had uninterrupted $PM_{10}$ time series.

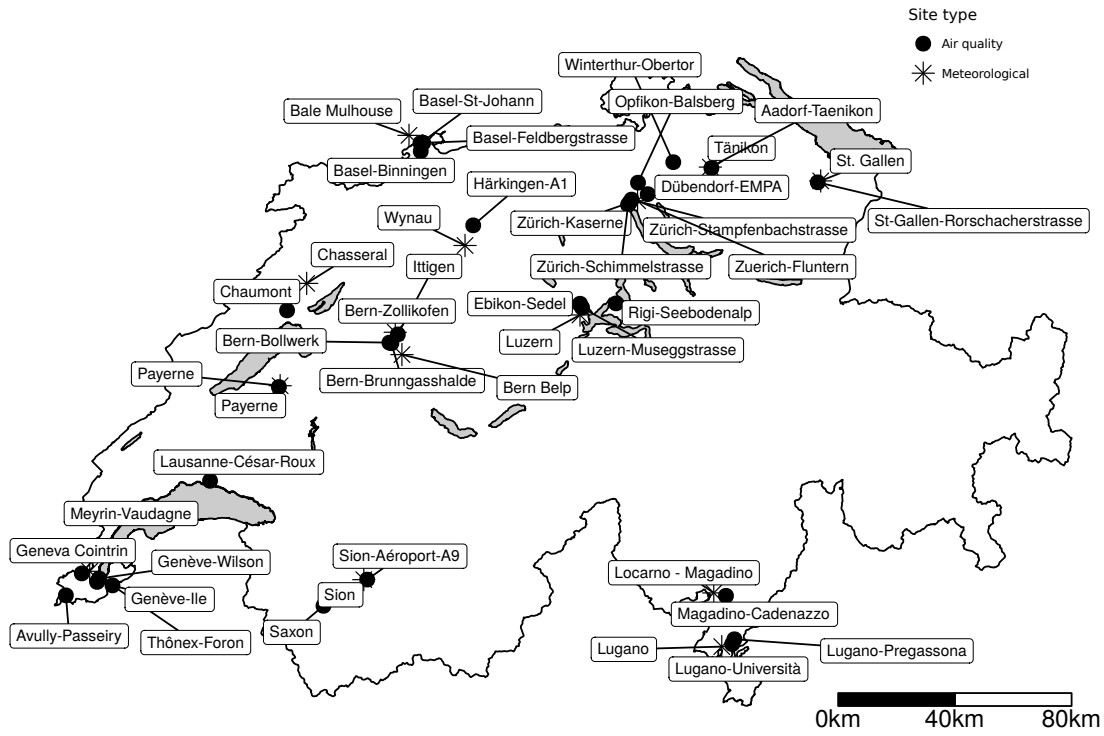

**Figure 2.** Locations of the air quality and meteorological sites included in the analysis. The map outline is the extent of Switzerland.

Surface meteorological variables to be included in the modelling process such as wind speed, wind direction, and atmospheric temperature were accessed from the Integrated Surface Database (ISD) with the **worldmet** R package (NOAA, 2016;



**Table 1.** Information for the PM$_{10}$ and meteorological monitoring sites used in this study.

| ID | Site name | Latitude | Longitude | Elevation (m) | Site type | Site name ISD (met.) | Data span |
|----|-----------|----------|-----------|---------------|-----------|----------------------|-----------|
| 1 | Avully-Passeiry | 46.163 | 6.005 | 427 | Rural | Geneva Cointrin | 2001–2016 |
| 2 | Magadino-Cadenazzo | 46.160 | 8.934 | 203 | Rural | Locarno - Magadino | 1997–2016 |
| 3 | Payerne | 46.813 | 6.944 | 489 | Rural | Payerne | 1997–2016 |
| 4 | Saxon | 46.139 | 7.148 | 460 | Rural | Sion | 1998–2016 |
| 5 | Tänikon | 47.480 | 8.905 | 538 | Rural | Aadorf-Taenikon | 2002–2016 |
| 6 | Härkingen-A1 | 47.312 | 7.821 | 431 | Rural motorway | Wynau | 1997–2016 |
| 7 | Sion-Aéroport-A9 | 46.220 | 7.342 | 483 | Rural motorway | Sion | 1997–2016 |
| 8 | Chaumont | 47.050 | 6.979 | 1136 | Rural mountain | Chasseral | 2002–2016 |
| 9 | Rigi-Seebodenalp | 47.067 | 8.463 | 1031 | Rural mountain | Luzern | 2002–2016 |
| 10 | Basel-Binningen | 47.541 | 7.583 | 316 | Suburban | Bale Mulhouse | 1997–2016 |
| 11 | Dübendorf-EMPA | 47.403 | 8.613 | 432 | Suburban | Zuerich-Fluntern | 1997–2016 |
| 12 | Ebikon-Sedel | 47.068 | 8.301 | 482 | Suburban | Luzern | 2002–2016 |
| 13 | Ittigen | 46.976 | 7.479 | 460 | Suburban | Bern-Zollikofen | 2002–2016 |
| 14 | Lugano-Pregassona | 46.026 | 8.968 | 305 | Suburban | Lugano | 2007–2016 |
| 15 | Meyrin-Vaudagne | 46.231 | 6.074 | 439 | Suburban | Geneva Cointrin | 2002–2016 |
| 16 | Opfikon-Balsberg | 47.439 | 8.570 | 430 | Suburban | Zuerich-Fluntern | 2001–2016 |
| 17 | Thônex-Foron | 46.196 | 6.211 | 422 | Suburban | Geneva Cointrin | 2002–2016 |
| 18 | Basel-St-Johann | 47.566 | 7.582 | 260 | Urban background | Bale Mulhouse | 1997–2016 |
| 19 | Lugano-Università | 46.011 | 8.957 | 280 | Urban background | Lugano | 1997–2016 |
| 20 | Luzern-Museggstrasse | 47.056 | 8.310 | 460 | Urban background | Luzern | 2002–2010 |
| 21 | Winterthur-Obertor | 47.500 | 8.732 | 448 | Urban background | Zuerich-Fluntern | 2000–2014 |
| 22 | Zürich-Kaserne | 47.378 | 8.530 | 409 | Urban background | Zuerich-Fluntern | 1997–2016 |
| 23 | Basel-Feldbergstrasse | 47.567 | 7.595 | 255 | Urban traffic | Bale Mulhouse | 2004–2016 |
| 24 | Bern-Bollwerk | 46.951 | 7.441 | 536 | Urban traffic | Bern Belp | 1997–2016 |
| 25 | Bern-Brunngasshalde | 46.949 | 7.450 | 533 | Urban traffic | Bern-Zollikofen | 2002–2015 |
| 26 | Genève-Ile | 46.206 | 6.143 | 375 | Urban traffic | Geneva Cointrin | 2001–2008 |
| 27 | Genève-Wilson | 46.216 | 6.151 | 376 | Urban traffic | Geneva Cointrin | 2002–2013 |
| 28 | Lausanne-César-Roux | 46.522 | 6.640 | 530 | Urban traffic | Geneva Cointrin | 1997–2016 |
| 29 | St-Gallen-Rorschacherstrasse | 47.429 | 9.387 | 660 | Urban traffic | St. Gallen | 2001–2013 |
| 30 | Zürich-Schimmelstrasse | 47.371 | 8.524 | 415 | Urban traffic | Zuerich-Fluntern | 1997–2016 |
| 31 | Zürich-Stampfenbachstrasse | 47.387 | 8.540 | 445 | Urban traffic | Zuerich-Fluntern | 1997–2016 |

Carslaw, 2017). These observations are generally available as hourly means and were therefore aggregated to daily averages. The wind speed aggregation used was the scalar averages which represents average atmospheric motion well at this aggregation period (Grange, 2014). Generally, the closest ISD site with a complete time series was matched to an air quality monitoring site, but there were cases where the data record was poor for the closest site, or it was unrepresentative (usually due to large

5 differences in elevation) so another ISD site was used instead. Some air quality monitoring sites monitor meteorological variables, but often the time series were not complete in the ISD database and another site was therefore supplemented. Fourteen unique ISD sites were used and Table 1 shows which ISD site was used for each of the 31 air quality monitoring sites.

Synoptic scale weather patterns were included into the models by using the Swiss Weather Type Classifications (WTC). The WTC is an objective and automatic classification scheme which is used to describe broad synoptic scale circulation patterns in

10 Switzerland. There are ten different WTCs types but only the CAP9 classification was used which defines nine distinct clusters





of synoptic weather patterns calculated by principal component analysis (Weusthoff, 2011). Descriptions of what these nine classes represent can be found in the supplementary material (Table A1).

Modelled daily boundary layer heights between 1997 and 2016 were sourced from the European Centre for Medium-Range Weather Forecasts (ECMWF) ERA-Interim data portal (Dee et al., 2011). The highest spatial resolution outputs were used which were at $0.125 \times 0.125$ decimal degrees. The NetCDF ECMWF model outputs were promoted to a raster stack and the midday boundary layer heights were extracted for each of the 31 monitoring sites (Hijmans, 2016; Pierce, 2017). Many of the Swiss urban monitoring sites are within close proximity and therefore only 23 unique raster cells were needed to represent the 31 sites. After the raster extraction, daily time series of boundary layer heights for each site were generated. The modelled ECMWF outputs were tested against radio sounding observations at Payerne before 2010 when such data exists. Although the two data sets did not agree well, a positive correlation was present and inclusion of boundary layer variable was done to allow the models to have a predictor which represented approximate atmospheric stability and the modelled data was judged to be suitable for this purpose.

For each of the 23 raster cells, daily back trajectories were calculated using the HYSPLIT model for the monitored period of $PM_{10}$ (1997–2016) (Stein et al., 2015). The back trajectories were calculated backwards in time for 120 hours and used half the mean monthly boundary layer height as their starting height. This start height ensured that the back trajectory receptor was aloft, but remained within the boundary layer throughout the year. The back trajectories were then clustered into six clusters using the Euclidian distance and these clusters were used to represent the common air masses the $PM_{10}$ monitoring sites were exposed to. The use of six clusters was a heuristic, but the six clusters represented distinct air masses and they were very stable across the 23 receptor locations. The HYSPLIT clustering function in **openair** was used to determine these clusters (Carslaw and Ropkins, 2012).

## 2.2 Modelling

RF models which used $PM_{10}$ as the dependent variable for each of the 31 air quality monitoring sites were grown. All RF models used the same explanatory variables to predict daily $PM_{10}$ concentrations. The explanatory variables were: wind speed, wind direction, atmospheric temperature, synoptic weather pattern, boundary layer height, air mass cluster based on the HYSPLIT back trajectories, a linear trend term which was the Unix time of the observation (number of seconds since 1 January, 1970), Julian day (day of the year) as the seasonal term, and day of the week. The air mass cluster, the synoptic weather pattern, and day of the week variables were categorical variables while all others were numeric. All variables were used within their response scale with no transformations being applied. The **randomForest** R package was used as the interface to the RF functions reported by Breiman (2001) (Liaw and Wiener, 2002). A daily $PM_{10}$ concentration was only modelled if valid wind speed data was available for that day. For all other input variables, missing data was imputed with the median of numeric variables and the mode for categorical variables. Training of the models was conducted on 80 % of the input data and the other 20 % was withheld from the training and used to validate the models once they had been grown.

RF only requires a handful of tuning parameters (also called hyper parameters) to be specified by the user (Liaw and Wiener, 2002; Immitzer et al., 2012). To determine the optimal values, many models were run with different combinations of tuning





parameters. The model performance statistics using the testing set (data withheld from the training step) and run times were evaluated to judge what hyper parameters grew the best performing models. For this application, the models were found to be rather insensitive to tuning parameters. However, the number of variables used to grow a tree was set to three, the minimum node-size or depth was five, and the number of trees within a forest was set at 300 for all models.

### 2.2.1 Meteorological normalisation

The meteorological normalisation of the daily $PM_{10}$ time series was achieved by repeatedly sampling and predicting using individual site RF models, rather than attempting to solve-for, and then remove the short term variation in a time series. The RF predictive model for a site was used to predict every $PM_{10}$ concentration 1000 times. For every prediction, the explanatory variables, with the exception of the trend term, were sampled without replacement and randomly allocated to a dependent variable observation (a $PM_{10}$ concentration). The 1000 predictions were then aggregated using the arithmetic mean and this represented "average" meteorological conditions and hence, this was the meteorologically normalised trend. If more than a thousand predictions were made, only a very minor reduction of noise was achieved. The functions used to grow the RF models and apply the meteorological normalisation procedure reported here are available in the **normalweatherr** R package (Grange, 2017).

### 2.3 Trend tests

After the normalised time series for a site had been calculated, formal trend tests were preformed. The Theil-Sen estimator accounting for autocorrelation was used at the 95 % confidence level ($\alpha = 0.05$) to indicate a significant trend. The autocorrelation consideration process results in more conservative confidence intervals for the trend estimates. These functions were also provided by the **openair** R package (Carslaw and Ropkins, 2012).

## 3 Results and discussion

### 3.1 Random forest model evaluation

The predictive random forest (RF) models performed well for most $PM_{10}$ monitoring sites. All mean squared errors (MSE) and $R^2$ values are displayed in tabular form in the supplementary material (Table A2). $R^2$ values ranged from 54 to 71 % (Figure 3). This indicates for some sites in Switzerland $PM_{10}$ concentrations could be well explained by a combination of surface meteorological conditions, boundary layer height, synoptic scale conditions, back trajectory receptor air mass clusters, and time variables which acted as proxies for emission strength. There were only two obvious patterns observed between site type and predictive model performance: the rural motorway sites performed in a similar way and the rural mountain sites' models generally performed worse than other site types when using the $R^2$ metric. However, there were only two of each of these site types analysed in this study, and the other four site types did not demonstrate any conclusive grouping with model performance measures (Figure 3).





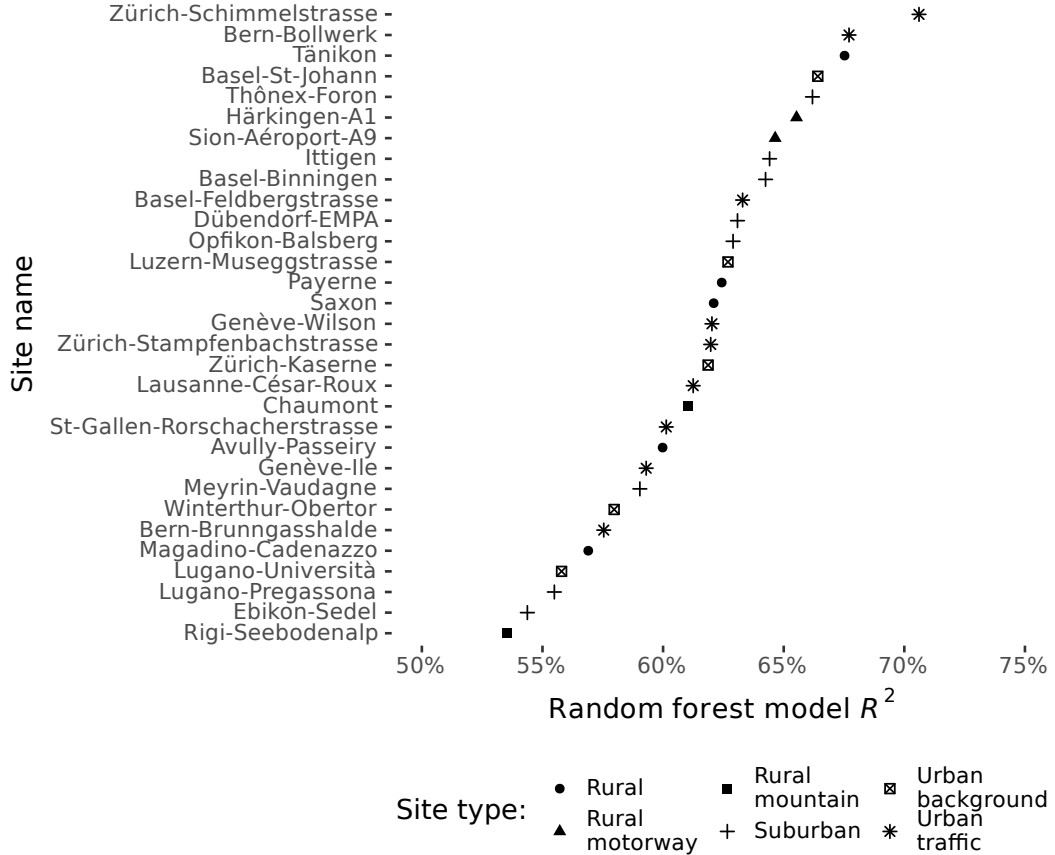

**Figure 3.** The $R^2$ values for the 31 random forest models grown for the Swiss $PM_{10}$ monitoring sites.

The most important explanatory variable for $PM_{10}$ concentrations depended on which site was being investigated. However, generally, wind speed was the variable with the greatest importance for prediction (Figure 4). Other sites demonstrated that the seasonal term (Julian day), or trajectory cluster were the most important variables to explain variability in $PM_{10}$ concentrations (Figure 4). This indicates that both local and regional scale processes were important when explaining $PM_{10}$ concentrations in

5    Switzerland. Day of the week and the synoptic-scale classification (WTC) were generally the least important variables in the RF models, but both variables always contributed to the models' predictive ability (Figure 4). Including variables with little predictive power does not negatively effect the performance of RF models and therefore there was no attempt to remove such variables from the models. Interestingly, wind direction was often a relatively unimportant variable (Figure 4). This may be due to daily wind direction averages not contributing much information gain in the model because the aggregation period results

10   in the metric representing atmospheric motion rather poorly. For all of the 31 sites, the normalised $PM_{10}$ was approximately monotonic and no seasonal component was apparent which made formal trend tests suitable.



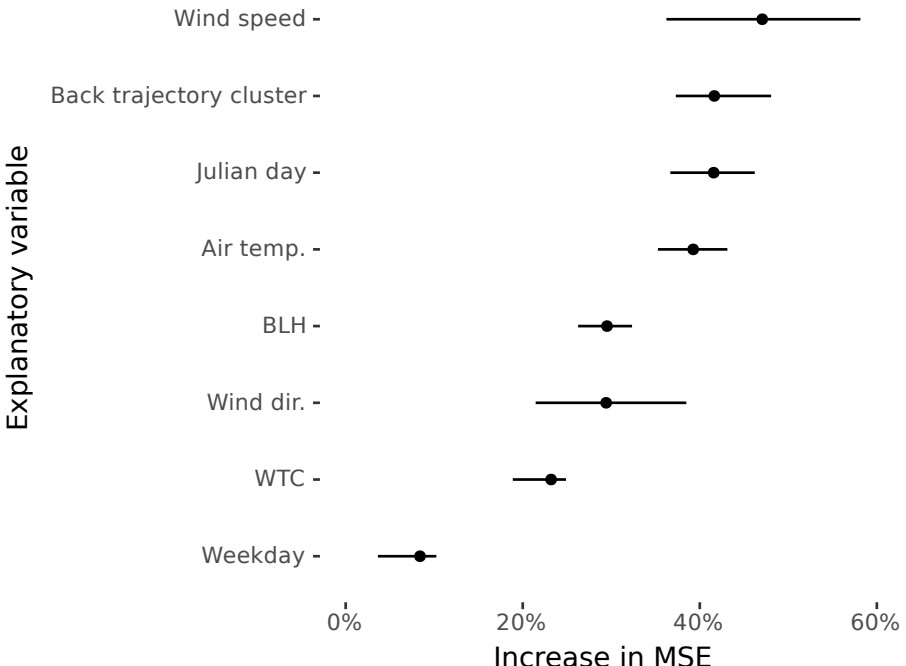

**Figure 4.** Variable importance for the 31 Swiss $PM_{10}$ monitoring sites' random forest models. Dots represent the mean increase in mean square error (MSE) and the lines represent the interquartile range for each variable.

## 3.2 $PM_{10}$ trend analysis

In all but two $PM_{10}$ Swiss monitoring sites, normalised $PM_{10}$ concentrations were found to be significantly decreasing at the $\alpha = 0.05$ level between 1997 and 2016. Significantly decreasing normalised $PM_{10}$ trends at individual sites ranged from -0.09 to -1.16 $\mu g\,m^{-3}\,year^{-1}$ (Figure A1). These values were similar to the normalised trends reported by Barmpadimos et al. (2011)

5   of -0.15 to -1.2 $\mu g\,m^{-3}\,year^{-1}$ which analysed Swiss $PM_{10}$ trends between 1991 and 2008 with a different method (general additive models; GAMs). The similarities between the two studies suggests that $PM_{10}$ concentrations have continued to reduce at the same rate as reported in the past, which also validates the performance of emission control measures and confirms the trends that were modelled based on emission inventories and their projections (Heldstab et al., 2013). Luzern-Museggstrasse was the only monitoring site which demonstrated a significantly increasing normalised $PM_{10}$ trend of 0.14 $\mu g\,m^{-3}\,year^{-1}$.

10   However, this facility stopped monitoring $PM_{10}$ in 2009 and therefore it is unknown if this trend continued to more recent times. The two monitoring sites in Geneva also did not have $PM_{10}$ observations to the end of the analysis period. $PM_{10}$ at Genève-Wilson demonstrated no significant normalised trend and Genève-Ile had the least significant normalised $PM_{10}$ trend across the 31 sites analysed (Figure A1). This may suggest that Geneva's $PM_{10}$ trends are different than the rest of Switzerland, but with the lack of more recent observations, this is uncertain.





Sites classified as 'urban traffic' had a greater decreasing trend when compared to other site types (Figure 5). When the six site type trends were aggregated together, the stronger decreasing trend for traffic sites was clear with an average trend of -0.77 $\mu g \, m^{-3} \, year^{-1}$, compared to the other site types which ranged between -0.39 and -0.63 $\mu g \, m^{-3} \, year^{-1}$ (Figure 5). Barmpadimos et al. (2011) also reported trends based on site type but their site type definitions were not the same as used in this study so they should not be directly compared. The higher first four points in the rural panel of Figure 5 was caused by the aggregated time series only containing the Magadino-Cadenazzo monitoring site at the very beginning of analysis period. Magadino-Cadenazzo is located south of the Alps and experiences higher average concentrations of $PM_{10}$ compared to the other rural sites. Without the observations from the other rural sites, these higher concentrations leveraged the mean seen in Figure 5. These observations were still included in the analysis and the Theil-Sen estimator used is hardened against outliers so this will have minimal influence on the trend estimate.

Difference in annual $PM_{10}$ mean concentrations between the rural and urban traffic site types for 2016, the final year of analysis, was 4.7 $\mu g \, m^{-3}$ compared to 9.8 $\mu g \, m^{-3}$ in 1997. The deltas between rural and other site types (excluding the mountainous sites) also decreased during the analysis period. This suggests the locations which are influenced by immediate $PM_{10}$ sources are becoming less polluted by local emissions and are increasingly heading towards rural background levels. The rural and urban background sites' trend metrics are very similar indicating that these two site types are behaving in a very similar way in respect to changes to $PM_{10}$ concentrations over time.

The site type classifications used in this study can be sorted by their increasing anthropogenic $PM_{10}$ load in this order: rural mountain, rural, suburban, urban background, and urban traffic. Site types which experience more anthropogenic $PM_{10}$ emissions could be expected to demonstrate greater reductions in $PM_{10}$ concentrations when emission inventions or controls are applied. This continuum is only partially shown in the trend magnitudes however with suburban and rural motorway sites not conforming to this expected pattern (Figure 5). In fact, the suburban sites demonstrate the smallest decrease in $PM_{10}$ concentrations.

The rural motorway trends can be explained because although PM (tailpipe) emissions for road traffic have decreased in Switzerland between 1997 and 2016, the volume of traffic using the adjacent roads has increased (Bundesamt für Strassen, 2017). This increase in traffic would have offset the lower emissions during the time period and thus PM concentrations would not have decreased as much as could be expected based on vehicular emissions alone. The suburban sites lack of decrease is more difficult to explain. There are many processes which could explain this feature, but we attribute this result due to changes in the surrounding environment of the suburban sites. Many of the monitoring sites in Switzerland which are classed as suburban have become increasingly urban during the period of analysis (1997 and 2016). Therefore, some of these suburban monitoring sites are being influenced by more urban-like processes and emissions due to the development in their vicinity.

The comparison of the RF meteorological normalisation models with other techniques was not a primary objective of this work. However, it is important to consider what effect meteorological normalisation had on the trend estimates. To investigate this, the $PM_{10}$ observations which were subjected to the meteorological normalisation process were aggregated to monthly means and their trends tested with the Theil-Sen test with identical parameters as used on the normalised time series. This could be considered a 'standard' and routine procedure for air quality data analysis. With the exception of the rural motorway sites,





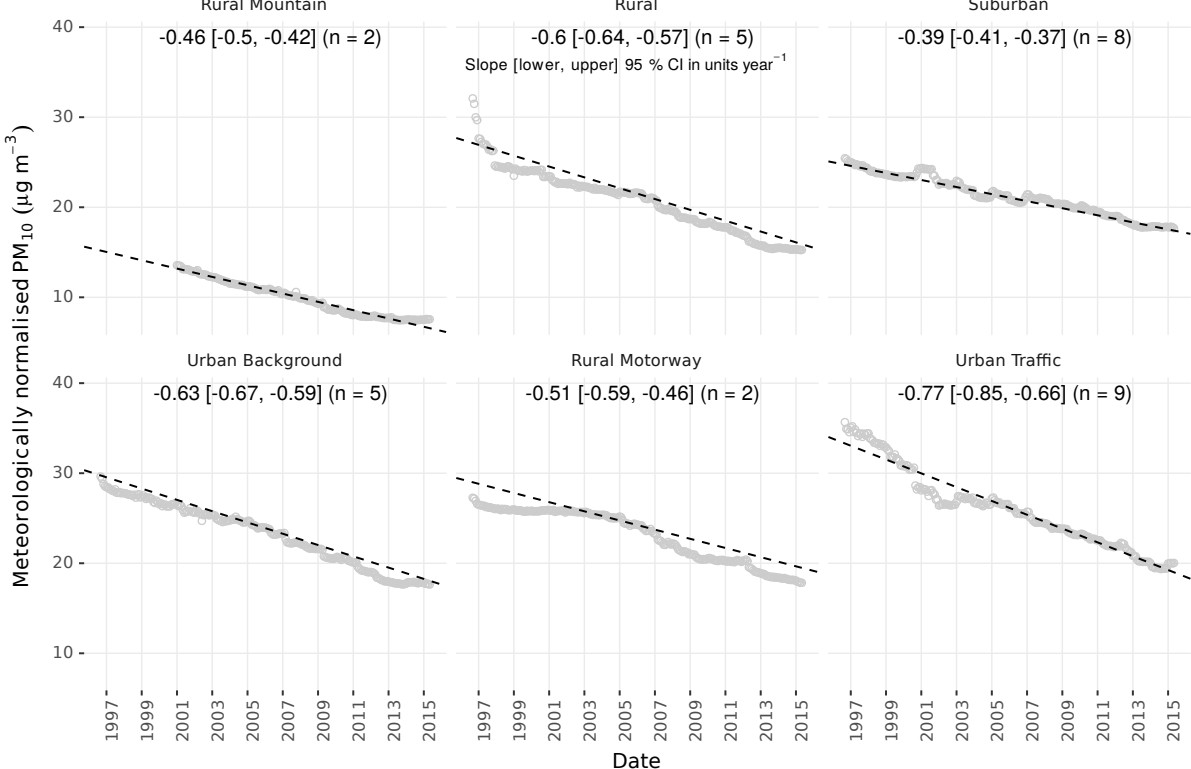

**Figure 5.** Aggregated meteorologically normalised $PM_{10}$ trends for the six site types in Switzerland between 1997 and 2016. Points represent the aggregated meteorologically normalised monthly means, lines represent the trend estimate, and $n$ represents the number of sites in the group.

the normalised trend estimate was found to be greater (more negative), than the non-normalised trend estimates (Figure 6). This indicates that meteorology in Switzerland between 1997 and 2016 has masked or obscured changes in $PM_{10}$ emissions during the same period in the observational record. Because the meteorological normalisation technique helps to explain variation in $PM_{10}$ concentrations, the normalised trend estimates had a much lower range of uncertainty when compared to the aggregated observations in all cases (Figure 6). Therefore, not only did the meteorological normalisation technique generally estimate more negative trends compared to standard methods, the trends calculated were more robust and less uncertain when compared to a routine analysis method which would lead to quicker identification of significant trends.

### 3.2.1 Explaining the observed trends

One of the primary advantages of decision tree methods like RF over other machine learning techniques is the ability to interpret and explain the models and discussion of this is presented in Section 1.3.1. Here, this advantage will be leveraged to help explain some of the features in the $PM_{10}$ trends in Switzerland between 1997 and 2016.



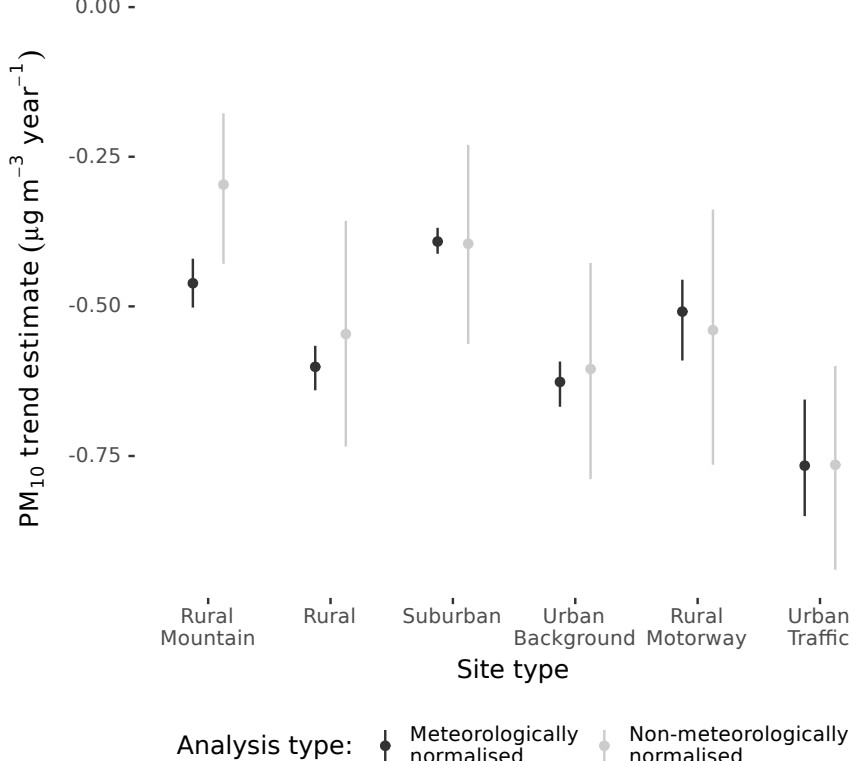

**Figure 6.** $PM_{10}$ trend slope estimates of meteorological normalised and non-meteorological normalised observations for five site types in Switzerland between 1997 and 2016. The line ranges represent the 95 % confidence intervals of the slope estimates.

Partial dependence plots allow RF models to be evaluated and to confirm how the explanatory variables are being used in the models for prediction (Jones and Linder, 2015). For the application presented here, there are general physical and chemical processes which should be confirmed in the RF models. For example, it can be expected that $PM_{10}$ concentrations will be inversely related to wind speed due to increased atmospheric dispersion, wintertime concentrations will be higher than other seasons resulting from a combination of greater emissions and atmospheric stability. These general predictions and processes were confirmed by the RF models' partial dependence plots (one site shown as an example in Figure 7).

The partial dependence plots of the Zürich-Stampfenbachstrasse RF model (Figure 7) showed some interesting features and were typical for Switzerland's traffic influenced sites. The $y$ (vertical) axes for each plot represents the dependence of $PM_{10}$ concentration on one variable if all other variables are fixed at their average level. The most important variable was wind speed at this location and the non-linear relationship is present in Figure 7. When wind speeds were very low, the $PM_{10}$ concentrations were on average over 38 $\mu g\,m^{-3}\,day^{-1}$ but the influence on $PM_{10}$ concentrations was strong and therefore at wind speeds greater than 3 $m\,s^{-1}$, average concentrations decreased to under 22 $\mu g\,m^{-3}\,day^{-1}$ (Figure 7). There was minimal evidence of



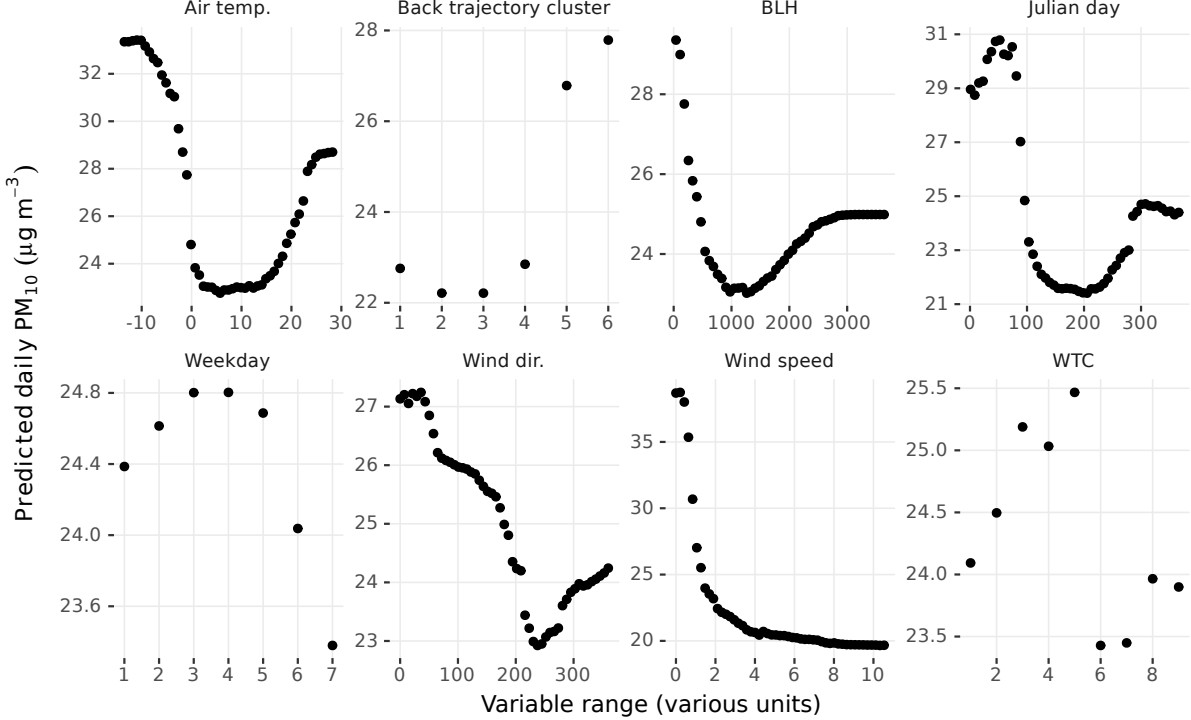

**Figure 7.** Partial dependence plots of the explanatory variables used in the Zürich-Stampfenbachstrasse PM$_{10}$ random forest model.

increasing PM$_{10}$ concentrations at high wind speeds due to resuspension of wind blown PM at any monitoring site in the RF models.

Weekday was the variable of least importance for the Zürich-Stampfenbachstrasse RF model but the partial dependence plot still demonstrates what would be expected. Weekdays (days 1–5; Monday–Friday) were more polluted than the weekend days

5 due to higher traffic sourced emissions, but the variability of PM$_{10}$ concentrations among the weekdays was less than 2 $\mu g\,m^{-3}$ day$^{-1}$, *i.e.*, the response scale was small (Figure 7). There was evidence of a sequential loading process over the weekdays which peaked on Thursdays (day 4) and also lower concentrations during the early working week (Monday and Tuesday; days 1 and 2) which resulted from reduced precursor PM emissions during the weekend, especially Sunday.

The seasonal component represented by Julian day showed a similar pattern to air temperature (Figure 7). Despite the similar

10 shapes of dependencies on PM$_{10}$ for these variables, they represent rather different processes. The Julian day dependence represents the changes in local and regional emissions which influence PM$_{10}$ concentrations over the course of the year. In the case of Zürich-Stampfenbachstrasse, this will be dominated by changes in regional background concentrations with the addition of local traffic emissions. The seasonal variation of emissions which effect PM$_{10}$ concentrations at Zürich-Stampfenbachstrasse spans 10 $\mu g\,m^{-3}$, and this indicates that the seasonal effect is important to consider. When Julian day was removed from the



RF models, the dependence on air temperature and boundary layer height did not change and this shows that the models were able to differentiate the different processes correctly despite their collinearity.

The back trajectory cluster variable was important for many $PM_{10}$ monitoring sites including Zürich-Stampfenbachstrasse (Figure 4 and 7). The decoded clusters' descriptions displayed in Figure 7 can be found in Table A3 but the two most polluted air masses, 5 and 6 represented a local flow from south west Switzerland and a strong north east flow from Poland and southern Germany respectively (Figure 8). This indicates that air masses from surrounding European states can cause polluted $PM_{10}$ conditions in Zürich, as can periods of calm and localised flows.

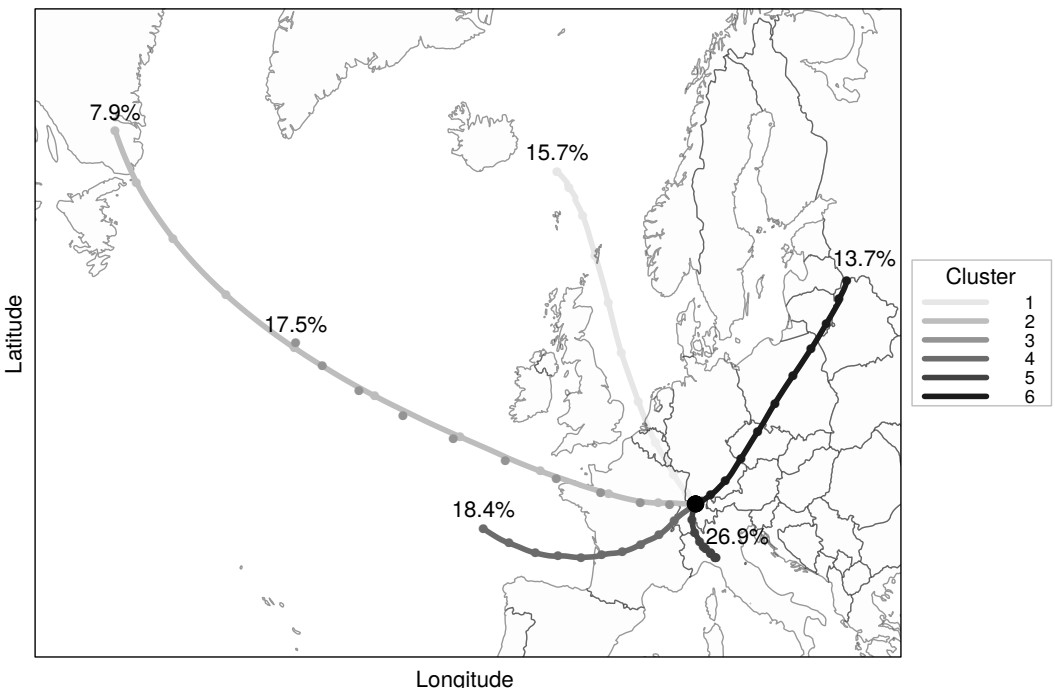

**Figure 8.** The six back trajectory clusters for the Zürich receptor location between 1997 and 2016 which were used by the random forest $PM_{10}$ models. The clusters are decoded in Table A3 and the percentages indicate the frequency of occurrence.

The partial dependence plots indicate that most monitoring sites experience their minimum $PM_{10}$ concentrations when the boundary layer is $\approx 1000$ metres high, but concentrations increase again once the boundary layer increases over 2000 metres (Figure 7). This is an interesting phenomenon and it suggests that there are two regimes in Switzerland which drive elevated $PM_{10}$ concentrations. The first is the obvious (and expected) combination of low temperatures, low boundary heights, and high rates of surface-based emissions during wintertime. These factors combine to create a poor dispersive environment which leads to high pollutant concentrations. The second regime which causes elevated $PM_{10}$ concentrations is active when temperatures are above $20\,^{\circ}\mathrm{C}$ and the boundary layer is above 2000 metres (Figure 7). These conditions occur with every air





mass cluster and under all synoptic weather patterns which are experienced at these higher temperatures. Therefore, this regime is associated with warm, dry, dusty, and deep convective boundary layer conditions which favour transportation of $PM_{10}$ from other locations and the generation of secondary aerosol and other processes driven by photochemistry. Daily sulphur (in $PM_{10}$) observations are available at the Payerne monitoring site and $SO_4$ concentrations do indeed increase at higher boundary layer

5  heights while primary pollutants such as $NO_x$ do not (Figure 9). These results are consistent with enhanced sulphate formation in summertime when the formation of sulphate through photochemistry is most important. By contrast, the concentration of primary pollutants such as $NO_x$ tend to decrease with increasing boundary layer height due to increased mixing.

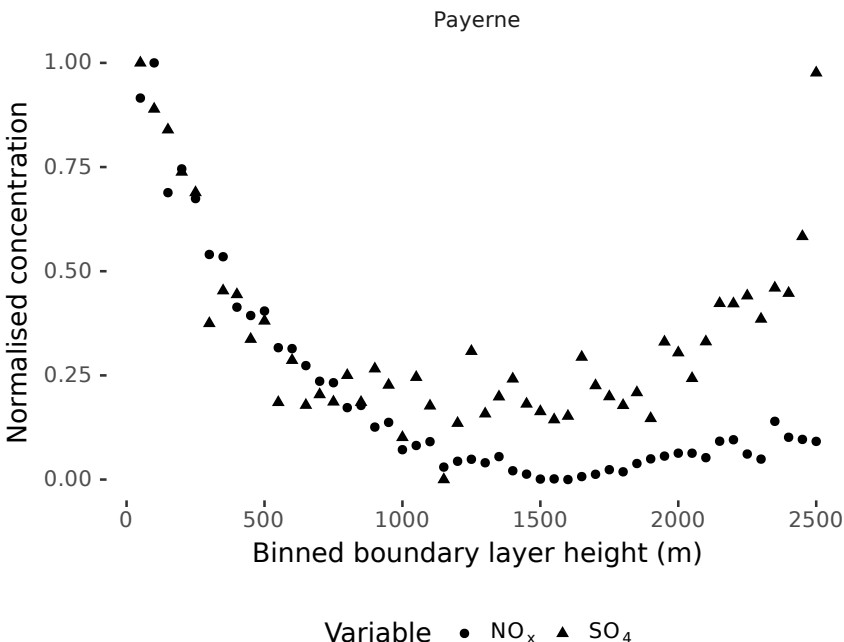

**Figure 9.** Mean normalised concentrations of $SO_4$, a secondary PM species and $NO_x$ for binned boundary layer heights (bin was set at 50 metres) at Payerne between 1997 and 2016.

The partial dependence plots of the seasonal and trend components also demonstrate that while the trend component decreased between 1997 and 2016, the seasonal component also decreased at some of the Swiss $PM_{10}$ monitoring sites. The best

10  example of this was demonstrated at Magadino-Cadenazzo, a rural site in Ticino in the south of Switzerland (Table 1 and Figure 2). The decrease in the seasonal component was especially true after 2006 and during early winter at Magadino-Cadenazzo (December; Figure 10(a)). As discussed above, this further validates air pollutant emission controls and interventions because both the background concentration and the local loading of $PM_{10}$ during winter is decreasing simultaneously. There is evidence however that the wintertime loading has plateaued since approximately 2014 at this monitoring site (Figure 10(b)).





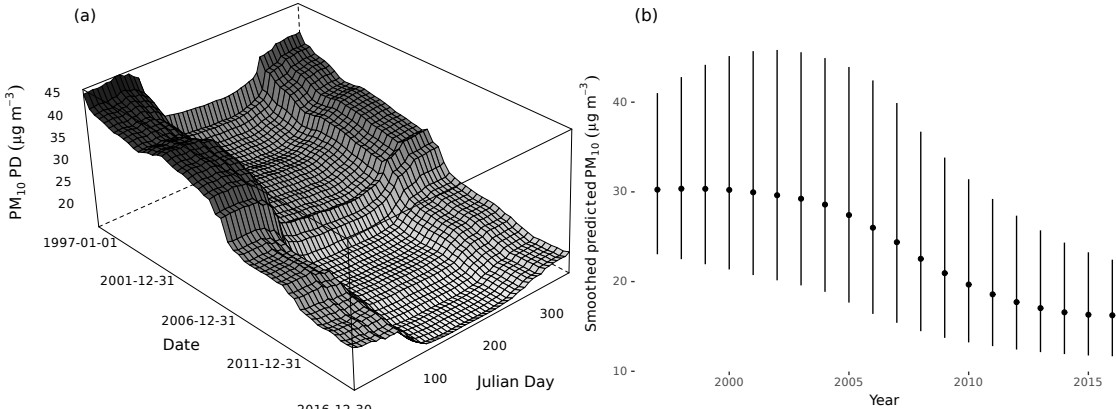

**Figure 10.** (a) $PM_{10}$ partial dependence on trend and seasonal components (Date and Julian day respectively) and (b) annual predicted seasonal component at Magadino-Cadenazzo where dots represent the mean and lines indicate the amplitude of the seasonal component.

The rural mountain Chaumont and Rigi-Seebodenalp monitoring sites have low $PM_{10}$ concentrations when compared to the other Swiss sites and site types (Figure A1 and Figure 5). Both of these locations are isolated and are located above 1000 metres of elevation (Table 1 and Figure 2). Therefore, these two monitoring sites represent pristine locations. The $PM_{10}$ concentrations at both locations decreased at $\approx$ -0.45 $\mu g\,m^{-3}\,year^{-1}$ between 1997 and 2016 indicating a wider-scale European reduction in

$PM_{10}$ and its precursors (Guerreiro et al., 2014). Interestingly, the normalised trend at Rigi-Seebodenalp showed an additional $PM_{10}$ loading between April 8 and 26, 2010 due to the Eyjafjallajökull Icelandic volcanic eruption (Bukowiecki et al., 2011; Thorsteinsson et al., 2012) but at Chaumont, this was not discernible (not shown). This demonstrates that the two sites do behave differently and are exposed to different processes at times. The differences between the two sites are not clear in the concentration data alone and demonstrates a potentially useful side effect of the technique where it can be used to investigate

abnormal events.

The RF models for these two rural and mountainous locations also demonstrated different processes compared to other site types. The most interesting feature was that the relationship between air temperature and boundary layer height with $PM_{10}$ concentrations differed from the other Swiss monitoring sites. The two mountainous sites experienced their highest $PM_{10}$ concentrations at high temperatures (Chaumont shown in Figure 11(a)). This difference in dependence was due to these

monitoring locations being intermittently above the boundary layer, which was also confirmed with the boundary layer height partial dependence plots (Figure 11(b)). When these elevated sites were within the boundary layer during warmer periods, the relatively well mixed $PM_{10}$ influenced the monitoring locations, but during cooler times, the sites were located in the free troposphere decoupled from surface based emissions. This generally resulted in the elevated monitoring sites experiencing lower concentrations of $PM_{10}$ during cooler periods which was not the case for monitoring sites located at lower elevations,

for example, Basel-St-Johann, an urban background site located at 260 metres of elevation (Figure 11).





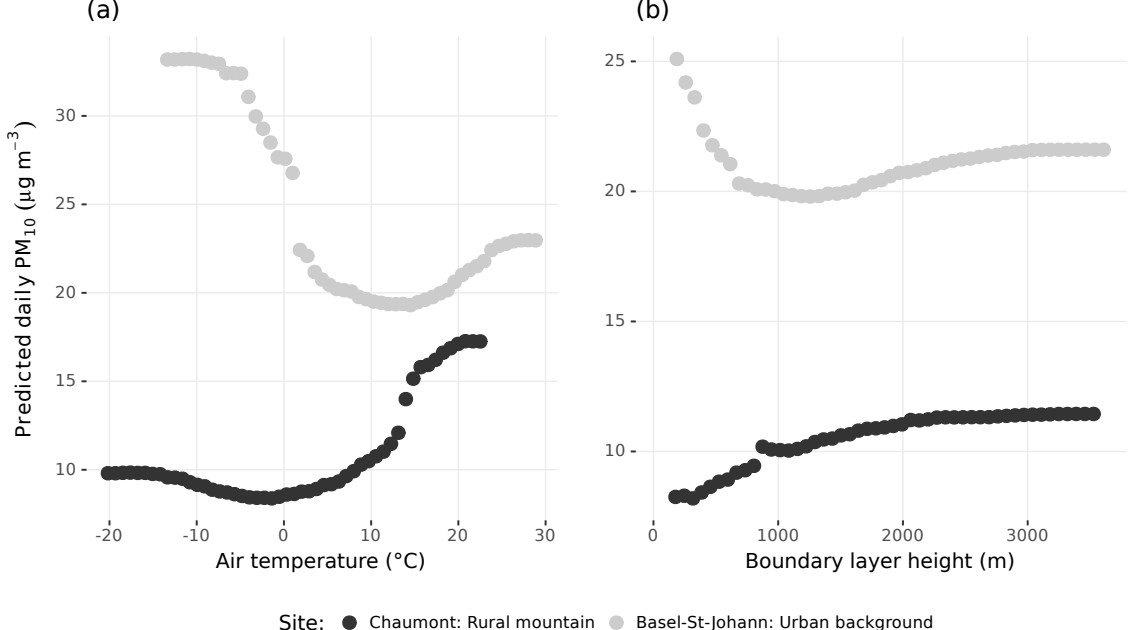

**Figure 11.** Partial dependence of $PM_{10}$ concentrations on (a) air temperature and (b) boundary layer height at two monitoring sites with different site type classifications.

## 4 Conclusions

This paper presented a meteorological normalised $PM_{10}$ trend analysis using daily data from Switzerland. Random forest (RF) predictive models which were used to explain variation of $PM_{10}$ concentrations using surface meteorology, synoptic scale weather patterns, boundary layer height, back trajectory clusters, and time variables. The models were then used to prepare the
$PM_{10}$ time series to create a meteorological normalised trend which was suitable for formal trend analysis.

The RF performed well for the 31 monitoring sites with $R^2$ values up to 71 %. Wind speed, Julian day (the seasonal component), and back trajectory cluster were generally the most important predictors for $PM_{10}$ concentration. For 29 of the 31 monitoring sites analysed, $PM_{10}$ concentrations were found to be significantly decreasing at rates between -0.09 and -1.16 $\mu g\,m^{-3}\,year^{-1}$ and on average, urban traffic sites demonstrated the greatest decrease of -0.77 $\mu g\,m^{-3}\,year^{-1}$. The RF
models' learning process was interpreted with partial dependence plots to explain the trends observed. There was evidence of a decrease in the seasonal component at some sites, *i.e.*, the wintertime loading has decreased, and the monitoring sites above 1000 metres of elevation showed interesting dependences on air temperature which were not demonstrated at other sites because they are intermittently located above the boundary layer. The models also indicated that across Switzerland, elevated $PM_{10}$ concentrations occur in poor dispersion conditions as well as at high temperatures with a deep boundary layers due to
high rates secondary PM generation resulting from photochemical processes.




The meteorological normalisation technique using RF was found to be helpful in the $PM_{10}$ trend analysis conducted and resulted in more negative and less uncertain trend estimates compared to another standard analysis method. The predictive modelling framework and technique was found to be easy to implement and user friendly because RF does not need to conform to strict parametric assumptions. The technique described could be used in many air quality exploratory data analysis applications.

*Code and data availability.* The data sources used in this work are described and referenced in the text. The code used to conduct the meteorological normalisation procedure is available as an open source R package (**normalweatherr**) and is also referenced in text.

*Competing interests.* The authors declare no competing interest.

*Acknowledgements.* S.K.G was supported by Anthony Wild with the provision of the Wild Fund Scholarship. S.K.G. also thanks Empa's Air Pollution/Environmental Technology research group for financial support while in Zürich. This work was partially funded by the NERC Air Pollution PhD studentships programme. The authors thank the University of York database administrators of Carl Stovell and his team and Rudolf Weber from the Swiss Federal Office for the Environment for the delivery of the Härkingen-A1 and Sion-Aéroport sites' monitoring data.



**Table A1.** The nine synoptic scale weather type classifications (WTC) used in this study (from Weusthoff, 2011).

| CAP9 class | CAP9 description |
|---|---|
| 1 | North-East, indifferent |
| 2 | West-South-West, cyclonic, flat pressure |
| 3 | Westerly flow over Northern Europe |
| 4 | East, indifferent |
| 5 | High Pressure over the Alps |
| 6 | North, cyclonic |
| 7 | West-South-West, cyclonic |
| 8 | High Pressure over Central Europe |
| 9 | Westerly flow over Southern Europe, cyclonic |





**Table A2.** Random forest model performance statistics for 31 $PM_{10}$ air quality monitoring sites in Switzerland.

| ID | Site | Site type | MSE | $R^2$ (%) |
|----|------|-----------|-----|-----------|
| 1 | Avully-Passeiry | Rural | 54.824 | 59.980 |
| 2 | Magadino-Cadenazzo | Rural | 129.356 | 56.898 |
| 3 | Payerne | Rural | 60.854 | 62.431 |
| 4 | Saxon | Rural | 64.023 | 62.097 |
| 5 | Tänikon | Rural | 51.140 | 67.523 |
| 6 | Härkingen-A1 | Rural motorway | 84.145 | 65.531 |
| 7 | Sion-Aéroport-A9 | Rural motorway | 53.355 | 64.646 |
| 8 | Chaumont | Rural mountain | 26.095 | 61.019 |
| 9 | Rigi-Seebodenalp | Rural mountain | 32.276 | 53.513 |
| 10 | Basel-Binningen | Suburban | 65.807 | 64.247 |
| 11 | Dübendorf-EMPA | Suburban | 64.563 | 63.084 |
| 12 | Ebikon-Sedel | Suburban | 68.702 | 54.373 |
| 13 | Ittigen | Suburban | 68.965 | 64.415 |
| 14 | Lugano-Pregassona | Suburban | 84.349 | 55.492 |
| 15 | Meyrin-Vaudagne | Suburban | 52.188 | 59.037 |
| 16 | Opfikon-Balsberg | Suburban | 57.011 | 62.900 |
| 17 | Thônex-Foron | Suburban | 61.899 | 66.192 |
| 18 | Basel-St-Johann | Urban background | 63.320 | 66.413 |
| 19 | Lugano-Università | Urban background | 173.909 | 55.792 |
| 20 | Luzern-Museggstrasse | Urban background | 89.484 | 62.690 |
| 21 | Winterthur-Obertor | Urban background | 68.498 | 57.971 |
| 22 | Zürich-Kaserne | Urban background | 73.583 | 61.867 |
| 23 | Basel-Feldbergstrasse | Urban traffic | 62.058 | 63.296 |
| 24 | Bern-Bollwerk | Urban traffic | 94.146 | 67.708 |
| 25 | Bern-Brunngasshalde | Urban traffic | 66.208 | 57.540 |
| 26 | Genève-Ile | Urban traffic | 66.777 | 59.299 |
| 27 | Genève-Wilson | Urban traffic | 80.017 | 62.025 |
| 28 | Lausanne-César-Roux | Urban traffic | 80.206 | 61.248 |
| 29 | St-Gallen-Rorschacherstrasse | Urban traffic | 55.139 | 60.131 |
| 30 | Zürich-Schimmelstrasse | Urban traffic | 91.317 | 70.609 |
| 31 | Zürich-Stampfenbachstrasse | Urban traffic | 75.976 | 61.974 |



**Table A3.** The six decoded HYSPLIT back trajectory clusters. The integer cluster key was used in the random forest models and the decoded cluster was determined after the cluster analysis.

| Cluster | Decoded cluster |
| --- | --- |
| 1 | Strong northerly flow from north sea |
| 2 | Very strong north west flow from Atlantic Ocean |
| 3 | Westerly flow from Atlantic Ocean |
| 4 | South west flow from France and western Switzerland |
| 5 | Local flow from south west Switzerland |
| 6 | Strong north east flow from Poland and southern Germany |





**Figure A1.** Meteorologically normalised PM$_{10}$ trends for the 31 sites analysed in Switzerland between 1997 and 2016.



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
