# Peer review of "Random forest meteorological normalisation models for Swiss $PM_{10}$ trend analysis"

_Atmospheric Chemistry and Physics, 2017_

## Referee Comment (RC1) · Anonymous Referee #2 · 20 Feb 2018

Scientific review comments

This paper describes the use of a random forest (RF) statistical approach to normalise long-term time series of PM10 measurements from 31 air quality monitoring sites in Switzerland. The advantage of the RF method used in this work is that it is possible to quantify, and hence interpret, the magnitude and significance of the explanatory variables. A number of putative explanatory variables are input into the RF model, but normalising for PM10 variability driven by variation in meteorology is the principal objective (and finding) of developing the normalisation routine.

The paper is a very well written description of the random forest statistical approach for meteorological normalisation of air quality time series, and of the interpretation of the findings of its application to time series of PM10 measurements in Switzerland. The

work is suitable for ACP and will be of international interest. I have no other scientific comments to make.

Editorial corrections

List of authors: present the superscripted numbers indicating affiliation in numerical order.

P3, L17: Insert 'of' to read '...of what is...'

P11, L13: Change 'than' to 'from'

P12, L6: Insert 'the' to read '...of the analysis period.'

P12, L11: Change to read '...in annual mean PM10 concentrations...'

P12, L26: Add apostrophe for "The suburban sites' lack of....'

P14, L4: Insert 'and that' before 'wintertime concentrations'

P14, L9: Rephrase to "The most important variable at this location was wind speed....'

P17, L4: Add the charge to the sulphate anion, i.e. SO42-

P17, caption to Fig. 9: Add the charge to the sulphate anion, i.e. SO42-

P26, L23: Is a URL available for the Fuller and Carslaw report "Putney High Street air quality, Part 2"?

---

## Referee Comment (RC2) · Anonymous Referee #1 · 16 Mar 2018

The paper describes a meteorological normalisation technique applied to time series of daily PM10 concentrations collected in 31 monitoring sites in Switzerland in the period 1997-2016. The technique is based on the Random-Forest (RF) model using various meteorological parameters (wind speed and direction, boundary layer height, weather pattern, etc.) as explanatory variables. Applying the proposed algorithm the meteorological effect on PM10 concentrations is removed and changes over time can be explained solely due to changes in emissions or chemistry. Also, RF presents the advantage of exploring the effect of each of the explanatory variables on controlling PM10 concentrations in relation to relevant physical and chemical processes through partial dependence plots.

The paper is well-written and relevant to the air quality community (and also to other

disciplines using time series).

Few comments that I would like to be addressed/discussed before final publication:

1. The authors found that PM10 concentrations in Switzerland decreased between 1997 and 2016 but the discussion about the rate of change is a bit vague. Authors point that similar trend rates were reported in Barmpadinos et al. (2011) between 1991 and 2008. I would expect greater decreasing trends in PM10 in light with recent technology developments in controlling PM emissions from diesel vehicles for instance.

2. What is the role of wood burning emissions in trends in PM10 concentrations in Switzerland? Does the authors have any estimate of the rate of use of wood burners in rural / urban areas? Do you think that wood burning emissions might have an effect on trends in PM10 in suburban areas?

3. Rural mountain sites are the ones that the RF model explained less variance (R2 < 63% based on Fig. 3). Might the low R2 score explain the difference in trend observed using normalised time series vs. non-normalised in Figure 6? The narrower confidence interval of the trends estimated based on normalised time series is due to the removal of variability but it does not inform about the "accuracy" of the trend value (i.e. the real trend value).

4. What it would be a good "R2" threshold to be confident that the RF model is repro-ducing "enough" variability of the original time series?

5. Is there any other advantage of using normalised time series to calculate trends than just obtaining more robust trend estimators? Based on Figure 6, trends estimates using normalised and non-normalised time series are the same within confidence intervals but the computational cost is higher. Might normalised time series be useful to explore step-changes in the time series and related to specific policy interventions?

6. Partial dependence plots. What is the advantage of using these plots rather than build them using the "raw" data?

---

## Author Comment (AC1) · 21 Apr 2018

Author responses to referee comments of acp-2017-1092

Editorial changes

1. List of authors: present the superscripted numbers indicating affiliation in numerical order.

The manuscript has been updated with this change.

Response to reviewers Anonymous Referee #1

The paper is well-written and relevant to the air quality community (and also to other disciplines using time series).

[Figure]

Thank you.

1. The authors found that PM10 concentrations in Switzerland decreased between 1997 and 2016 but the discussion about the rate of change is a bit vague. Authors point that similar trend rates were reported in Barmpadinos et al. (2011) between 1991 and 2008. I would expect greater decreasing trends in PM10 in light with recent technology developments in controlling PM emissions from diesel vehicles for instance.

The reduced PM emission from diesel vehicles during the past years is certainly one factor contributing to the observed and described downward trend of PM10 . A large fraction of PM exhaust emissions from diesel vehicles is elemental or equivalent black carbon (EC or eBC), so the absolute trend of PM exhaust emissions from diesel vehicles can roughly been estimated from the trend of EC or eBC. In a recent and yet unpublished study, the trend of eBC in Switzerland has been determined. It has been found that eBC is declining at urban traffic sites at a rate of about 0.2 $\mu$g m $-3$ year $-1$ , at urban and suburban sites the trend is smaller and about 0.1 $\mu$g m $-3$ year $-1$ . In terms of PM10, the improvements due to reductions of emissions of PM from the exhaust of diesel vehicles is small, however, the reductions are significant for the trend of eBC. This is in fact not surprising as primary particle emissions from the exhaust of road traffic is only one source among many other sources of PM10 and atmospheric PM10 formation processes (note that about 50% of PM10 in Switzerland as well as elsewhere is secondary aerosol formed from gaseous precursors). We can conclude that the observed trends of PM10 in Switzerland are clearly larger than the contribution that can be expected from reduced exhaust emissions from diesel vehicles. Therefore, more explanation for the drivers of the trends observed would be speculative. With this considered, we believe the manuscript does not need to be modified.

2. What is the role of wood burning emissions in trends in PM10 concentrations in Switzerland? Does the authors have any estimate of the rate of use of wood burners in rural/urban areas? Do you think that wood burning emissions might have an effect on trends in PM10 in suburban areas?

Wood burning is a significant source of PM10 in Switzerland, both in the alpine region but also in suburban and urban areas (see e.g. Zotter et al., Atmospheric Chemistry and Physics 2014; Herich et al., Atmospheric Environment, 2014). According to the Swiss Federal Office of Energy (SFOE), the number of wood burning appliances in Switzerland as well as the total heating power has been declining since 1990 (-17.2% and -18.6%, respectively) (see Schweizerische Holzenergiestatistik 2016, Swiss Federal Office of Energy, http://www.bfe.admin.ch/themen/00526/00541/00543/index.html?lang=en&dossier_id=00771). These changes in number and heating power are mainly due to the declining number of installed small wood stoves, the number and heating power of automated larger appliances is increasing. It can be assumed that the declining heating power and the shift from small stoves to larger automated appliances lead to lower total PM10 emissions from wood burning and that wood burning emissions contribute to the observed downward trends of PM10 in Switzerland. A targeted trend assessment for the contribution of wood burning emissions to PM10 is, however, not available. The number and heating power of wood burning appliances in Switzerland have not been differentiated for different regions or environments, it is therefore difficult to conclude if there is a particular effect on the trends in PM10 in suburban areas.

A paragraph has been added to the manuscript discussing this: "Woodburning is a source of PM10 in the alpine, suburban, and urban areas in Switzerland. The number of woodburning appliances and heating demand is deceasing over time and this change will contribute to the trends observed in Figure 5 (Stettler and Betbèze, 2017). However, the quantification of the reduction in woodburning activity on PM10 concentrations among the different site types is cannot be conducted with the current data concerning woodburner usage."

3. Rural mountain sites are the ones that the RF model explained less variance ($R^2 <$ 63 % based on Fig. 3). Might the low $R^2$ score explain the difference in trend observed using normalised time series vs. non-normalised in Figure 6? The narrower confidence

interval of the trends estimated based on normalised time series is due to the removal of variability but it does not inform about the "accuracy" of the trend value (i.e. the real trend value).

The rural mountain sites were indeed the sites with the lowest random forest performance, but we do not believe that the lack of performance explains the difference between the normalised and non-normalised trend estimates. There were only two rural mountain sites used in the analysis and therefore the greater difference between the meteorologically normalised and standard trend estimates can be explained as a function of the sample size. If the most closely related "rural" sites are sampled repeatedly to contain only two sites, and the standard trend analysis conducted, the slope estimates range from -0.29 to -0.75 $\mu$g m $-3$ year $-1$ (a delta of 0.46). In comparison the delta between the rural mountain sites' estimates was 0.17 $\mu$g m $-3$ year $-1$ . This indicates that the lack of sites of the rural mountain type can explain the greater difference of the trend estimates when compared to other site types and more sites will be needed to represent the Swiss rural mountain environments better. The lower performance of the rural mountain could be partially explained by the meteorological site selection being more important than other areas of less complicated terrain. The manuscript dies mention that there were only two rural mountain sites used in the analysis when discussing Figure 3.

4. What it would be a good "R2" threshold to be confident that the RF model is reproducing "enough" variability of the original time series?

The minimum for a "good" R2 value is a subjective measure and will vary for every application. As a rule, the minimum R2 value which would make a random forest model useful in the context of this technique will be about 50 %. If a random forest model explained less than 50 % of the dependent variable's variation, this would indicate there are physical and/or chemical processes which are not being represented by the explanatory variables which would be an interesting result in itself. In such cases, after training, there would not be much value in undertaking the sample-and-predict logic

to normalise the time series for average weather and other analyses would be more suitable.

5. Is there any other advantage of using normalised time series to calculate trends than just obtaining more robust trend estimators? Based on Figure 6, trends estimates using normalised and non-normalised time series are the same within confidence intervals but the computational cost is higher. Might normalised time series be useful to explore step-changes in the time series and related to specific policy interventions?

This is a very good suggestion and a research project is currently under-way to show how the meteorological normalisation technique can be used to robustly investigate interventions. Here, we narrowed our objectives to a trend analysis study and therefore intervention exploration was outside the scope of the current work.

6. Partial dependence plots. What is the advantage of using these plots rather than build them using the "raw" data?

Investigating the dependencies between pollutant concentrations and meteorological variables is a valid and useful technique. The advantage of partial dependence plots is that they control for variation of other variables and therefore isolate the contribution of the variable more effectively. For this application, the partial dependence plots offer a critical advantage of interpretability in how the random forest models are using the variables for learning prediction. This legibility is important to interpret the models' suitability for this particular application.

Anonymous Referee #2

The work is suitable for ACP and will be of international interest. I have no other scientific comments to make.

Thank you.

1. P3, L17: Insert 'of' to read '...of what is...'

The manuscript has been updated with this change.

2. P11, L13: Change 'than' to 'from'

The manuscript has been updated with this change.

3. P12, L6: Insert 'the' to read '...of the analysis period.'

The manuscript has been updated with this change.

4. P12, L11: Change to read '...in annual mean PM10 concentrations...'

The manuscript has been updated with this change.

5. P12, L26: Add apostrophe for 'The suburban sites' lack of...'

The manuscript has been updated with this change.

6. P14, L4: Insert 'and that' before 'wintertime concentrations'.

The manuscript has been updated with this change.

7. P14, L9: Rephrase to 'The most important variable at this location was wind speed...'

The manuscript has been updated with this change.

8. P17, L4: Add the charge to the sulphate anion, i.e. $SO^{2-}_4$

The manuscript has been updated with this change.

9. P17, caption to Fig. 9: Add the charge to the sulphate anion, i.e. $SO^{2-}_4$

The manuscript has been updated with this change. The figure legend has also been edited to contain the charge of sulphate.

10. P26, L23: Is a URL available for the Fuller and Carslaw report "Putney High Street air quality, Part 2"?

This report does not have a current URL but we can provide a copy if a request is made

to the corresponding author.